# Improving Sample Efficiency in Model-Free Reinforcement Learning from Images

## Abstract

Training an agent to solve control tasks directly from high-dimensional images with model-free reinforcement learning (RL) has proven difficult. The agent needs to learn a latent representation together with a control policy to perform the task. Fitting a high-capacity encoder using a scarce reward signal is not only sample inefficient, but also prone to suboptimal convergence. Two ways to improve sample efficiency are to extract relevant features for the task and use off-policy algorithms. We dissect various approaches of learning good latent features, and conclude that the image reconstruction loss is the essential ingredient that enables efficient and stable representation learning in image-based RL. Following these findings, we devise an off-policy actor-critic algorithm with an auxiliary decoder that trains end-to-end and matches state-of-the-art performance across both model-free and model-based algorithms on many challenging control tasks. We release our code to encourage future research on image-based RL[1].

## 1 Introduction

Cameras are a convenient and inexpensive way to acquire state information, especially in complex, unstructured environments, where effective control requires access to the proprioceptive state of the underlying dynamics. Thus, having effective RL approaches that can utilize pixels as input would potentially enable solutions for a wide range of real world problems.

The challenge is to efficiently learn a mapping from pixels to an appropriate representation for control using only a sparse reward signal. Although deep convolutional encoders can learn good representations (upon which a policy can be trained), they require large amounts of training data. As existing reinforcement learning approaches already have poor sample complexity, this makes direct use of pixel-based inputs prohibitively slow. For example, model-free methods on Atari (Bellemare et al., 2013) and DeepMind Control (DMC) (Tassa et al., 2018) take tens of millions of steps (Mnih et al., 2013; Barth-Maron et al., 2018), which is impractical in many applications, especially robotics.

A natural solution is to add an auxiliary task with an unsupervised objective to improve sample efficiency. The simplest option is an autoencoder with a pixel reconstruction objective. Prior work has attempted to learn state representations from pixels with autoencoders, utilizing a two-step training procedure, where the representation is first trained via the autoencoder, and then either with a policy learned on top of the fixed representation (Lange & Riedmiller, 2010; Munk et al., 2016; Higgins et al., 2017b; Zhang et al., 2018; Nair et al., 2018), or with planning (Mattner et al., 2012; Finn et al., 2015). This allows for additional stability in optimization by circumventing dueling training objectives but leads to suboptimal policies. Other work utilizes end-to-end model-free learning with an auxiliary reconstruction signal in an on-policy manner (Jaderberg et al., 2017).

We revisit the concept of adding an autoencoder to model-free RL approaches, but with a focus on off-policy algorithms. We perform a sequence of careful experiments to understand why previous approaches did not work well. We found that a pixel reconstruction loss is vital for learning a good representation, specifically when trained end-to-end. Based on these findings, we propose a simple autoencoder-based off-policy method that *can be trained end-to-end*. Our method is the first model-free off-policy algorithm to successfully train simultaneously both the latent state representation and policy in a stable and sample-efficient manner.

---

[1] An anonymous website with code, results, and videos: https://sites.google.com/view/sac-ae/home

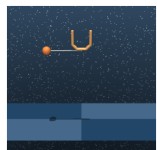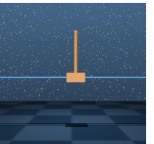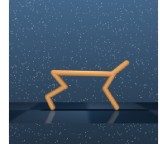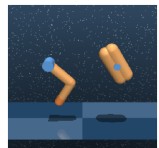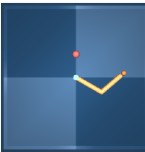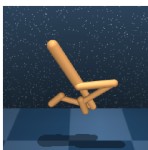

Figure 1: Image-based continuous control tasks from the DeepMind control suite (Tassa et al., 2018) used in our experiments. Each task offers an unique set of challenges, including complex dynamics, sparse rewards, hard exploration, and more. Refer to Appendix A for more information.

Of course, some recent state-of-the-art model-based RL methods (Hafner et al., 2018; Lee et al., 2019) have demonstrated superior sample efficiency to leading model-free approaches on pixel tasks from (Tassa et al., 2018). But we find that our model-free, off-policy, autoencoder-based approach is able to match their performance, closing the gap between model-based and model-free approaches in image-based RL, despite being a far simpler method that does not require a world model.

This paper makes three main contributions: (i) a demonstration that adding a simple auxiliary reconstruction loss to a model-free off-policy RL algorithm achieves comparable results to state-of-the-art model-based methods on the suite of continuous control tasks from Tassa et al. (2018); (ii) an understanding of the issues involved with combining autoencoders with model-free RL in the off-policy setting that guides our algorithm; and (iii) an open-source PyTorch implementation of our simple method for researchers and practitioners to use as a strong baseline that may easily be built upon.

## 2 RELATED WORK

Efficient learning from high-dimensional pixel observations has been a problem of paramount importance for model-free RL. While some impressive progress has been made applying model-free RL to domains with simple dynamics and discrete action spaces (Mnih et al., 2013), attempts to scale these approaches to complex continuous control environments have largely been unsuccessful, both in simulation and the real world. A glaring issue is that the RL signal is much sparser than in supervised learning, which leads to sample inefficiency, and higher dimensional observation spaces such as pixels worsens this problem.

One approach to alleviate this problem is by training with auxiliary losses. Early work (Lange & Riedmiller, 2010) explores using deep autoencoders to learn feature spaces in visual reinforcement learning, crucially Lange & Riedmiller (2010) propose to recompute features for all collected experiences after each update of the autoencoder, rendering this approach impractical to scale to more complicated domains. Moreover, this method has been only demonstrated on toy problems. Alternatively, Finn et al. (2015) apply deep autoencoder pretraining to real world robots that does not require iterative re-training, improving upon computational complexity of earlier methods. However, in this work the linear policy is trained separately from the autoencoder, which we find to not perform as well as end-to-end methods.

Shelhamer et al. (2016) use auxiliary losses in Atari that incorporate forward and inverse dynamics with A3C, an on-policy algorithm. They recommend a multi-task setting and learning dynamics and reward to find a good representation, which relies on the assumption that the dynamics in the task are easy to learn and useful for learning a good policy. Jaderberg et al. (2017) propose to use unsupervised auxiliary tasks, both observation-based and reward-based based off of real world inductive priors, and show improvements in Atari, again in the on-policy regime, which is much more stable for learning. Unfortunately, this work also relies on inductive biases by designing internal rewards to learn a good representation which is hard to scale to the real world problems. Higgins et al. (2017b); Nair et al. (2018) use a beta variational autoencoder ($\beta$-VAE) (Kingma & Welling, 2013; Higgins et al., 2017a) and attempt to extend unsupervised representation pretraining to the off-policy setting, but find it hard to perform end-to-end training, thus receding to the iterative re-training procedure (Lange & Riedmiller, 2010; Finn et al., 2015).

There has been more success in using model-based methods on images, such as Hafner et al. (2018); Lee et al. (2019). These methods use a world model (Ha & Schmidhuber, 2018) approach, learning a representation space using a latent dynamics loss and pixel decoder loss to ground on the original observation space. These model-based reinforcement learning methods often show improved sample efficiency, but with the additional complexity of balancing various auxiliary losses, such as a dynamics loss, reward loss, and decoder loss in addition to the original policy and value optimiza-

tions. These proposed methods are correspondingly brittle to hyperparameter settings, and difficult to reproduce, as they balance multiple training objectives.

To close the gap between model-based and model-free image-based RL in terms of sample efficiency and sidestep the issues of model learning, our goal is to train a model-free off-policy algorithm with auxiliary reconstruction loss in a stable manner.

## 3 BACKGROUND

A fully observable Markov decision process (MDP) is described by tuple $\langle \mathcal{S}, \mathcal{A}, P, R, \gamma \rangle$, where $\mathcal{S}$ is the state space, $\mathcal{A}$ is the action space, $P(\mathbf{s}_{t+1}|\mathbf{s}_t, \mathbf{a}_t)$ is the probability distribution over transitions, $R(\mathbf{s}_t, \mathbf{a}_t, \mathbf{s}_{t+1})$ is the reward function, and $\gamma$ is the discount factor (Bellman, 1957). An agent starts in a initial state $\mathbf{s}_1$ sampled from a fixed distribution $p(\mathbf{s}_1)$, then at each timestep $t$ it takes an action $\mathbf{a}_t \in \mathcal{A}$ from a state $\mathbf{s}_t \in \mathcal{S}$ and moves to a next state $\mathbf{s}_{t+1} \sim P(\cdot|\mathbf{s}_t, \mathbf{a}_t)$. After each action the agent receives a reward $r_t = R(\mathbf{s}_t, \mathbf{a}_t, \mathbf{s}_{t+1})$. We consider episodic environments with the length fixed to $T$. The goal of standard RL is to learn a policy $\pi(\mathbf{a}_t|\mathbf{s}_t)$ that can maximize the agent's expected cumulative reward $\sum_{t=1}^{T} \mathbb{E}_{(\mathbf{s}_t, \mathbf{a}_t) \sim \rho_\pi}[r_t]$, where $\rho_\pi$ is a state-action marginal distribution induced by the policy $\pi(\mathbf{a}_t|\mathbf{s}_t)$ and transition distribution $P(\mathbf{s}_{t+1}|\mathbf{s}_t, \mathbf{a}_t)$. An important modification (Ziebart et al., 2008) augments this objective with an entropy term $\mathcal{H}(\pi(\cdot|\mathbf{s}_t))$ to encourage exploration and robustness to noise. The resulting maximum entropy objective is then defined as:

$$\pi^* = \arg\max_\pi \sum_{t=1}^{T} \mathbb{E}_{(\mathbf{s}_t, \mathbf{a}_t) \sim \rho_\pi}[r_t + \alpha\mathcal{H}(\pi(\cdot|\mathbf{s}_t))],$$

where $\alpha$ is a temperature parameter that balances between optimizing for the reward and for the stochasticity of the policy.

We build on Soft Actor-Critic (SAC) (Haarnoja et al., 2018), an *off-policy* actor-critic method that uses the maximum entropy framework to derive soft policy iteration. At each iteration SAC performs a soft policy evaluation step and a soft policy improvement step. The soft policy evaluation step fits a parametric soft Q-function $Q(\mathbf{s}_t, \mathbf{a}_t)$ (critic) by minimizing the soft Bellman residual:

$$J(Q) = \mathbb{E}_{(\mathbf{s}_t, \mathbf{a}_t, r_t, \mathbf{s}_{t+1}) \sim \mathcal{D}}\left[\left(Q(\mathbf{s}_t, \mathbf{a}_t) - r_t - \gamma\mathbb{E}_{\mathbf{a}_{t+1} \sim \pi}\left[\bar{Q}(\mathbf{s}_{t+1}, \mathbf{a}_{t+1}) - \alpha\log\pi(\mathbf{a}_{t+1}|\mathbf{s}_{t+1})\right]\right)^2\right],$$

$$\tag{1}$$

where $\mathcal{D}$ is the replay buffer, and $\bar{Q}$ is the target soft Q-function parametrized by a weight vector obtained using the exponentially moving average of the soft Q-function weights to stabilize training. The soft policy improvement step then attempts to learn a parametric policy $\pi(\mathbf{a}_t|\mathbf{s}_t)$ (actor) by directly minimizing the KL divergence between the policy and a Boltzmann distribution induced by the current soft Q-function, producing the following objective:

$$J(\pi) = \mathbb{E}_{\mathbf{s}_t \sim \mathcal{D}}\left[\mathbb{E}_{\mathbf{a}_t \sim \pi}[\alpha\log(\pi(\mathbf{a}_t|\mathbf{s}_t)) - Q(\mathbf{s}_t, \mathbf{a}_t)]\right]. \tag{2}$$

The policy $\pi(\mathbf{a}_t|\mathbf{s}_t)$ is parametrized as a diagonal Gaussian to handle continuous action spaces.

When learning from raw images, we deal with the problem of partial observability, which is formalized by a partially observable MDP (POMDP). In this setting, instead of getting a low-dimensional state $\mathbf{s}_t \in \mathcal{S}$ at time $t$, the agent receives a high-dimensional observation $\mathbf{o}_t \in \mathcal{O}$, which is a rendering of potentially incomplete view of the corresponding state $\mathbf{s}_t$ of the environment (Kaelbling et al., 1998). This complicates applying RL as the agent now needs to also learn a compact latent representation to infer the state. Fitting a high-capacity encoder using only a scarce reward signal is sample inefficient and prone to suboptimal convergence. Following prior work (Lange & Riedmiller, 2010; Finn et al., 2015) we explore unsupervised pretraining via an image-based autoencoder. In practice, the autoencoder is represented as a convolutional encoder $f_{\mathrm{enc}}$ that maps an image observation $\mathbf{o}_t$ to a low-dimensional latent vector $\mathbf{z}_t$, and a deconvolutional decoder $f_{\mathrm{dec}}$ that reconstructs $\mathbf{z}_t$ back to the original image $\mathbf{o}_t$. The optimization is done by minimizing the standard reconstruction objective:

$$J(\mathrm{AE}) = \mathbb{E}_{\mathbf{o}_t \sim \mathcal{D}}\left[\frac{1}{2}||f_{\mathrm{dec}}(\mathbf{z}_t) - \mathbf{o}_t||_2^2\right] \quad \text{where} \quad \mathbf{z}_t = f_{\mathrm{enc}}(\mathbf{o}_t). \tag{3}$$

Or in the case of $\beta$-VAE (Kingma & Welling, 2013; Higgins et al., 2017a), where the variational distribution is parametrized as diagonal Gaussian, the objective is defined as:

$$J(\text{VAE}) = \mathbb{E}_{\mathbf{o}_t \sim \mathcal{D}} \Big[ \frac{1}{2} ||f_{\text{dec}}(\mathbf{z}_t) - \mathbf{o}_t||_2^2 - \frac{\beta}{2} \sum_i (1 + \log \boldsymbol{\sigma}_{t,i}^2 - \mathbf{z}_{t,i}^2 - \boldsymbol{\sigma}_{t,i}^2) \Big], \qquad (4)$$

where $\mathbf{z}_t = f_{\text{enc}}(\mathbf{o}_t)$ and $\boldsymbol{\sigma}_t^2 = f_{\text{enc\_std}}(\mathbf{o}_t)$. The latent vector $\mathbf{z}_t$ is then used by an RL algorithm, such as SAC, instead of the unavailable true state $\mathbf{s}_t$. To infer temporal statistics, such as velocity and acceleration, it is common practice to stack three consecutive frames to form a single observation (Mnih et al., 2013). We emphasize that in contrast to model-based methods (Ha & Schmidhuber, 2018; Hafner et al., 2018), we do not predict future states and solely focus on learning representations from the current observation to stay model-free.

## 4 A DISSECTION OF LEARNING STATE REPRESENTATIONS WITH $\beta$-VAE

In this section we explore in a systematic fashion how model-free *off-policy* RL can be made to train directly from pixel observations. We start by noting a dramatic performance drop when SAC is trained on pixels instead of proprioceptive state (Section 4.2) in the off-policy regime. This result motivates us to explore different ways of employing auxiliary supervision to speed up representation learning. While a wide range of auxiliary objectives could be added to aid effective representation learning, for simplicity we focus our attention on autoencoders. We follow Lange & Riedmiller (2010); Finn et al. (2015) and in Section 4.3 try an iterative unsupervised pretraining of an autoencoder that reconstructs pixels and is parameterized by $\beta$-VAE as per Nair et al. (2018); Higgins et al. (2017a). Exploring the training procedure used in previous work shows it to be sub-optimal and points towards the need for end-to-end training of the $\beta$-VAE with the policy network. Our investigation in Section 4.4 renders this approach useless due to severe instability in training, especially with larger $\beta$ values. We resolve this by using deterministic forms of the variational autoencoder (Ghosh et al., 2019) and a careful learning procedure. This leads to our algorithm, which is described and evaluated in Section 5.

### 4.1 EXPERIMENTAL SETUP

We briefly state our setup here, for more details refer to Appendix B. Throughout the paper we evaluate on 6 image-based challenging continuous control tasks from Tassa et al. (2018) depicted in Figure 1. For a concise presentation, in some places of the main paper we choose to plot results for reacher_easy, ball_in_cup_catch, and walker_walk only, while full results are available in the Appendix. An episode for each task results in maximum total reward of 1000 and lasts for exactly 1000 steps. Image observations are represented as $3 \times 84 \times 84$ RGB renderings, where each pixel is scaled down to $[0, 1]$ range. To infer velocity and acceleration we stack 3 consecutive frames following standard practice from Mnih et al. (2013). We keep the hyper parameters fixed across all tasks, except for action repeat, which we set only when learning from pixels according to Hafner et al. (2018) for a fair comparison to the baselines. If action repeat is used, the number of training observations is only a fraction of the environment steps (e.g. a 1000 steps episode at action repeat 4 will only result in 250 training observations). The exact action repeat settings can be found in Appendix B.3. We evaluate an agent after every 10000 training observation, by computing an average total reward across 10 evaluation episodes. For reliable comparison we run 10 random seeds for each configuration and compute mean and standard deviation of the evaluation reward.

### 4.2 MODEL-FREE OFF-POLICY RL WITH NO AUXILIARY TASKS

We start with an experiment comparing a model-free and off-policy algorithm SAC (Haarnoja et al., 2018) on pixels, with two state-of-the-art model-based algorithms, PlaNet (Hafner et al., 2018) and SLAC (Lee et al., 2019), and an upper bound of SAC on proprioceptive state (Table 1). We see a large gap between the capability of SAC on pixels (SAC:pixel), versus PlaNet and SLAC, which make use of many auxiliary tasks to learn a better representation, and can achieve performance close to the upper bound of SAC on proprioceptive state (SAC:state). From now, SAC:pixel will be our lower bound on performance as we gradually introduce different auxiliary reconstruction losses in order to close the performance gap.

| Task name | Number of Episodes | SAC:pixel | PlaNet | SLAC | SAC:state |
|---|---|---|---|---|---|
| `finger_spin` | 1000 | $645 \pm 37$ | $659 \pm 45$ | $\mathbf{900 \pm 39}$ | $\mathbf{945 \pm 19}$ |
| `walker_walk` | 1000 | $33 \pm 2$ | $949 \pm 9$ | $864 \pm 35$ | $\mathbf{974 \pm 1}$ |
| `ball_in_cup_catch` | 2000 | $593 \pm 84$ | $861 \pm 80$ | $932 \pm 14$ | $\mathbf{981 \pm 1}$ |
| `cartpole_swingup` | 2000 | $758 \pm 58$ | $802 \pm 19$ | - | $\mathbf{860 \pm 8}$ |
| `reacher_easy` | 2500 | $121 \pm 28$ | $\mathbf{949 \pm 25}$ | - | $\mathbf{953 \pm 11}$ |
| `cheetah_run` | 3000 | $366 \pm 68$ | $701 \pm 6$ | $830 \pm 32$ | $\mathbf{836 \pm 105}$ |

Table 1: A comparison over 6 DMC tasks of SAC from pixels, PlaNet, SLAC, and an upper bound of SAC from proprioceptive state, numbers are averaged over the last 5 episodes across 10 seeds. The large performance gap between SAC:pixel and SAC:state motivates us to address the representation learning bottleneck in model-free off-policy RL. Best performance bolded.

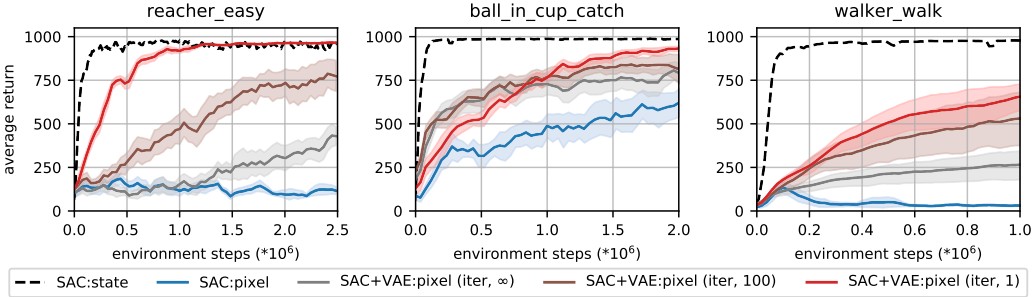

Figure 2: Separate $\beta$-VAE and policy training with no shared gradients SAC+VAE:pixel (iter, $N$), with SAC:state shown as an upper bound. $N$ refers to frequency in environment steps at which the $\beta$-VAE updates after initial pretraining. More frequent updates are beneficial for learning better representations, but cannot fully address the gap in performance. Full results in Appendix C.

### 4.3 ITERATIVE REPRESENTATION LEARNING WITH $\beta$-VAE

Following Lange & Riedmiller (2010); Finn et al. (2015), we experiment with unsupervised representation pretraining using a pixel autoencoder, which speeds up representation learning in image-based RL. Taking into account successful results from Nair et al. (2018); Higgins et al. (2017b) of using a $\beta$-VAE (Kingma & Welling, 2013; Higgins et al., 2017a) in the iterative re-training setup, we choose to employ a $\beta$-VAE likewise. We then proceed to first learn a representation space by pretraining the $f_{\mathrm{enc}}$, $f_{\mathrm{enc\_std}}$, and $f_{\mathrm{dec}}$ networks of the $\beta$-VAE according to the loss $J(\mathrm{VAE})$ Equation (4) on data collected from a random policy. We then learn a control policy on top of the frozen latent representations $\mathbf{z}_t = f_{\mathrm{enc}}(\mathbf{o}_t)$. We tune $\beta$ for best performance, and find large $\beta$ to be worse, and that very small $\beta \in [10^{-8}, 10^{-6}]$ performed best. In Figure 2 we vary the frequency $N$ at which the representation space is updated, from $N = \infty$, where the representation is never updated after an initial pretraining period with randomly collected data, to $N = 1$ where the representation is updated after every policy update. There is a positive correlation between this frequency and the final policy performance. We emphasize that the gradients are never shared between the $\beta$-VAE for learning the representation space, and the actor-critic learning the policy. These results suggest that if we can combine the representation pretraining via a $\beta$-VAE together with the policy learning in a stable end-to-end procedure, we would expect better performance. However, we note that prior work (Nair et al., 2018; Higgins et al., 2017a) has been unable to successfully demonstrate this. Regardless, we next perform such experiment to gain better understanding on what goes wrong.

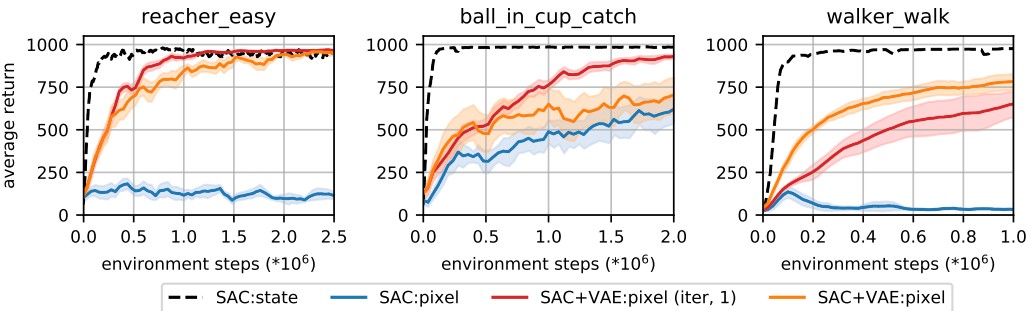

Figure 3: An unsuccessful attempt to propagate gradients from the actor-critic down to the encoder of the $\beta$-VAE to enable end-to-end off-policy training. The learning process of SAC+VAE:pixel exhibits instability together with the subpar performance comparing to the baseline SAC+VAE:pixel (iter, 1), which does not share gradients with the actor-critic. Full results in Appendix D.

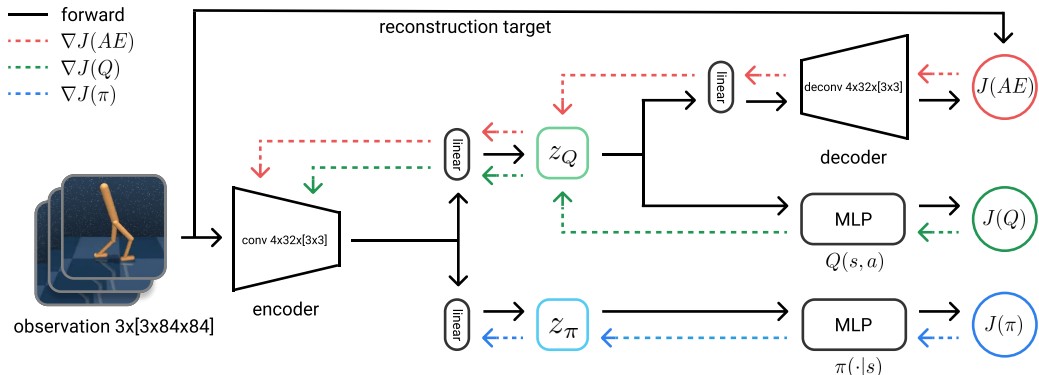

Figure 4: Our algorithm (SAC+AE) augments SAC (Haarnoja et al., 2018) with a regularized autoencoder (Ghosh et al., 2019) to achieve stable end-to-end training from images in the off-policy regime. The stability comes from switching to a deterministic encoder that is carefully updated with gradients from the reconstruction $J(\text{AE})$ (Equation (3)) and soft Q-learning $J(Q)$ (Equation (1)) objectives.

### 4.4 AN ATTEMPT FOR END-TO-END REPRESENTATION LEARNING WITH $\beta$-VAE

Our findings and the results from Jaderberg et al. (2017) motivate us to allow gradient propagation to the encoder of the $\beta$-VAE from the actor-critic, which in our case is SAC. We enable end-to-end learning by allowing the encoder to not only update with gradients from the $J(\text{VAE})$ loss (Equation (4), as done in Section 4.3, but also with gradients coming from the $J(Q)$ and $J(\pi)$ (Equations (1) and (2)) losses specified in Section 3. Results in Figure 3 show that the end-to-end policy learning together with the $\beta$-VAE in unstable in the off-policy setting and prone to divergent behaviours that hurt performance. Our conclusion supports the findings from Nair et al. (2018); Higgins et al. (2017a), which alleviate the problem by receding to the iterative re-training procedure. We next attempt stabilizing end-to-end training and introduce our method.

## 5 OUR METHOD: SAC+AE WITH END-TO-END OFF-POLICY TRAINING

We now seek to design a stable training procedure that can update the pixel autoencoder simultaneously with policy learning. We build on top of SAC (Haarnoja et al., 2018), a model-free and off-policy actor-critic algorithm. Based on our findings from Section 4, we propose a new, simple algorithm, SAC+AE, that enables end-to-end training. We notice that electing to learn deterministic latent representations, rather than stochastic as in the $\beta$-VAE case, has a stabilizing effect on the end-to-end learning in the off-policy regime. We thus use a deterministic autoencoder in a form of the regularized autoencoder (RAE) (Ghosh et al., 2019), that has many structural similarities with $\beta$-VAE. We also found it is important to update the convolutional weights in the target critic network faster, than the rest of the parameters. This allows faster learning while preserving the stability of the off-policy actor-critic. Finally, we share the encoder's convolutional weights between the actor and critic networks, but prevent the actor from updating them. Our algorithm is presented in Figure 4 for visual guidance.

### 5.1 PERFORMANCE ON PIXELS

We now show that our simple method, SAC+AE, achieves stable end-to-end training of an off-policy algorithm from images with an auxiliary reconstruction loss. We test our method on 6 challenging image-based continuous control tasks (see Figure 1) from DMC (Tassa et al., 2018). The RAE consists of a convolutional and deconvolutional trunk of 4 layers of 32 filters each, with $3 \times 3$ kernel size. The actor and critic networks are 3 layer MLPs with `ReLU` activations and hidden size of 1024. We update the RAE and actor-critic network at each environment step with a batch of experience sampled from a replay buffer. A comprehensive overview of other hyper paremeters is Appendix B.

We perform comparisons against several state-of-the-art model-free and model-based RL algorithms for learning from pixels. In particular: **D4PG** (Barth-Maron et al., 2018), an off-policy actor-critic algorithm, **PlaNet** (Hafner et al., 2018), a model-based method that learns a dynamics model with deterministic and stochastic latent variables and employs cross-entropy planning for control, and **SLAC** (Lee et al., 2019), which combines a purely stochastic latent model together with an model-free soft actor-critic. In addition, we compare against SAC that learns from low-dimensional pro-

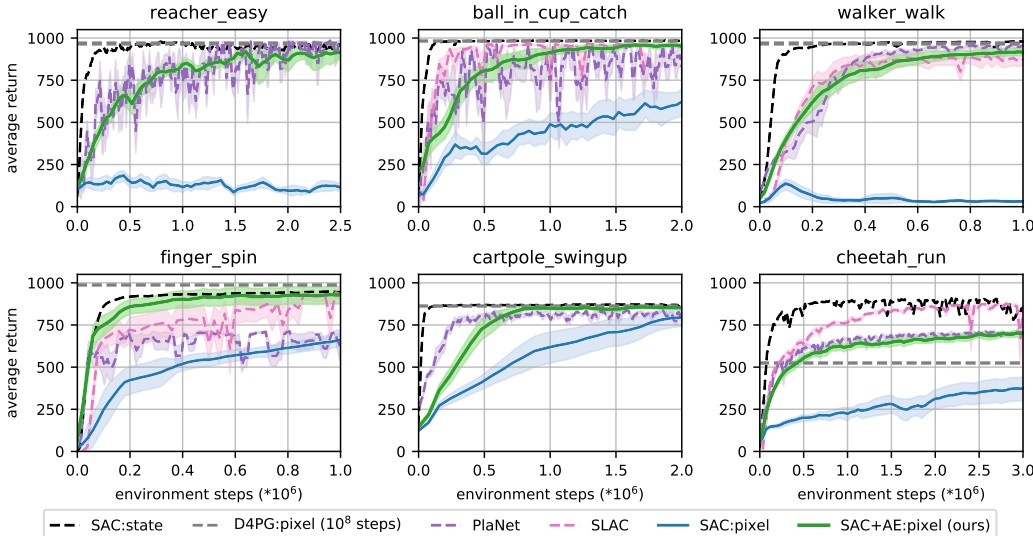

Figure 5: The **main result** of our work. Our method demonstrates significantly improved performance over the baseline SAC:pixel. Moreover, it matches the state-of-the-art performance of model-based algorithms, such as PlaNet (Hafner et al., 2018) and SLAC (Lee et al., 2019), as well as a model-free algorithm D4PG (Barth-Maron et al., 2018), that also learns from raw images. Our algorithm exhibits stable learning across ten random seeds and is extremely easy to implement.

prioceptive state, as an upper bound on performance. In Figure 5 we show that SAC+AE:pixel is able to match the state-of-the-art model-based methods such as PlaNet and SLAC, and significantly improve performance over the baseline SAC:pixel. Note that we use 10 random seeds, as recommended in Henderson et al. (2018) whereas the PlaNet and SLAC numbers shown are only over 4 and 2 seeds, respectively, as per the original publications.

## 6 ABLATIONS

To shed more light on some properties of the latent representation space learned by our algorithm we conduct several ablation studies. In particular, we want to answer the following questions: (i) is our method able to extract a sufficient amount of information from raw images to recover corresponding proprioceptive states readily? (ii) can our learned latent representation generalize to unseen tasks with similar image observations, but different reward objective, without reconstruction signal? Below, we answer these questions.

### 6.1 REPRESENTATION POWER OF THE ENCODER

Given how significantly our method outperforms a variant that does not have access to the image reconstruction signal, we hypothesize that the learned representation space encodes a sufficient amount of information about the internal state of the environment from raw images. Moreover, this information can be easily extracted from the latent state. To test this conjecture, we train SAC+AE:pixel and SAC:pixel until convergence on cheetah_run, then fix their encoders. We then train two

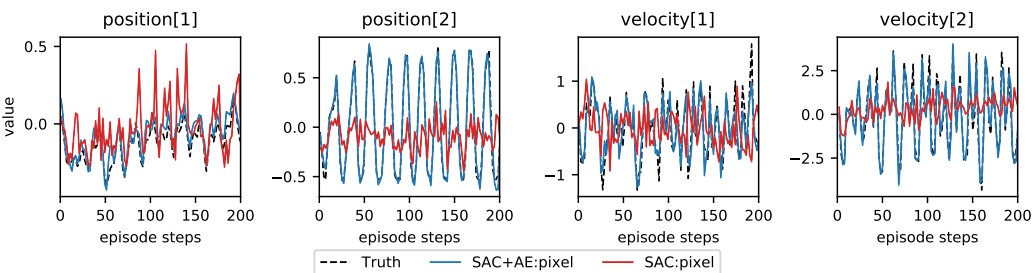

Figure 6: Linear projections of latent representation spaces learned by our method (SAC+AE:pixel) and the baseline (SAC:pixel) onto proprioceptive states. We compare ground truth value of each proprioceptive coordinate against their reconstructions for cheetah_run, and conclude that our method successfully encodes proprioceptive state information. For visual clarity we only plot 2 position (out of 8) and 2 velocity (out of 9) coordinates. Full results in Appendix F.

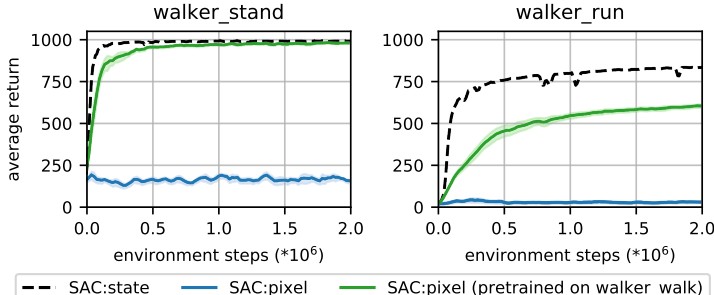

Figure 7: Encoder pretrained with our method (SAC+AE:pixel) on `walker_walk` is able to generalize to unseen `walker_stand` and `walker_run` tasks. All three tasks share similar image observations, but have quite different reward structure. SAC with a pretrained on `walker_walk` encoder achieves impressive final performance, while the baseline struggles to solve the tasks.

identical linear projections to map the encoders' latent embedding of image observations into the corresponding proprioceptive states. Finally, we compare ground truth proprioceptive states against their reconstructions on a sample episode. Results in Figure 6 confirm our hypothesis that the encoder grounded on pixel observations is powerful enough to almost perfectly restore the internals of the task, whereas SAC without the reconstruction loss cannot. Full results in Appendix F.

### 6.2 GENERALIZATION TO UNSEEN TASKS

To verify whether the latent representation space learned by our method is able to generalize to different tasks without additional fine-tuning with the reconstruction signal, we take three tasks `walker_stand`, `walker_walk`, and `walker_run` from DMC, which share similar observational appearance, but have different reward structure. We train an agent using our method (SAC+AE:pixel) on `walker_walk` task until convergence and extract its encoder. Consequently, we train two SAC agents *without reconstruction loss* on `walker_stand` and `walker_run` tasks from pixels. The encoder of the first agent is initialized with weights from the pretrained `walker_walk` encoder, while the encoder of the second agent is not. Neither of the agents use the reconstruction signal, and only backpropogate gradients from the critic to the encoder (see Figure 4). Results in Figure 7 suggest that our method learns latent representations that can readily generalize to unseen tasks and help a SAC agent achieve strong performance and solve the tasks.

## 7 DISCUSSION

We have presented the first end-to-end, off-policy, model-free RL algorithm for pixel observations with only reconstruction loss as an auxiliary task. It is competitive with state-of-the-art model-based methods, but much simpler, robust, and without requiring learning a dynamics model. We show through ablations the superiority of end-to-end learning over previous methods that use a two-step training procedure with separated gradients, the necessity of a pixel reconstruction loss over reconstruction to lower-dimensional "correct" representations, and demonstrations of the representation power and generalization ability of our learned representation.

We find that deterministic models outperform $\beta$-VAEs (Higgins et al., 2017a), likely due to the other introduced instabilities, such as bootstrapping, off-policy data, and end-to-end training with auxiliary losses. We hypothesize that deterministic models that perform better even in stochastic environments should be chosen over stochastic ones with the potential to learn probability distributions, and argue that determinism has the benefit of added interpretability, through handling of simpler distributions.

In the Appendix we provide results across all experiments on the full suite of 6 tasks chosen from DMC (Appendix A), and the full set of hyperparameters used in Appendix B. There are also additional experiments autoencoder capacity (Appendix E), a look at optimality of the learned latent representation (Appendix H), importance of action repeat (Appendix I), and a set of benchmarks on learning from proprioceptive observation (Appendix J). Finally, we opensource our codebase for the community to spur future research in image-based RL.

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

APPENDIX

## A  THE DEEPMIND CONTROL SUITE

We evaluate the algorithms in the paper on the DeepMind control suite (DMC) (Tassa et al., 2018) – a collection of continuous control tasks that offers an excellent testbed for reinforcement learning agents. The software emphasizes the importance of having a standardised set of benchmarks with a unified reward structure in order to measure made progress reliably.

Specifically, we consider six domains (see Figure 8) that result in twelve different control tasks. Each task (Table 2) poses a particular set of challenges to a learning algorithm. The `ball_in_cup_catch` task only provides the agent with a sparse reward when the ball is caught; the `cheetah_run` task offers high dimensional internal state and action spaces; the `reacher_hard` task requires the agent to explore the environment. We refer the reader to the original paper to find more information about the benchmarks.

| Task name | dim($\mathcal{O}$) | | dim($\mathcal{A}$) | Reward type |
|---|---|---|---|---|
| | Proprioceptive | Image-based | | |
| `ball_in_cup_catch` | 8 | $3 \times 84 \times 84$ | 2 | sparse |
| `cartpole_{balance,swingup}` | 5 | $3 \times 84 \times 84$ | 1 | dense |
| `cheetah_run` | 17 | $3 \times 84 \times 84$ | 6 | dense |
| `finger_{spin,turn_easy,turn_hard}` | 12 | $3 \times 84 \times 84$ | 2 | dense/sparse |
| `reacher_{easy,hard}` | 7 | $3 \times 84 \times 84$ | 2 | sparse |
| `walker_{stand,walk,run}` | 24 | $3 \times 84 \times 84$ | 6 | dense |

Table 2: We list specifications of observation space $\mathcal{O}$ (proprioceptive and image-based), action space $\mathcal{A}$, and the reward type for each task.

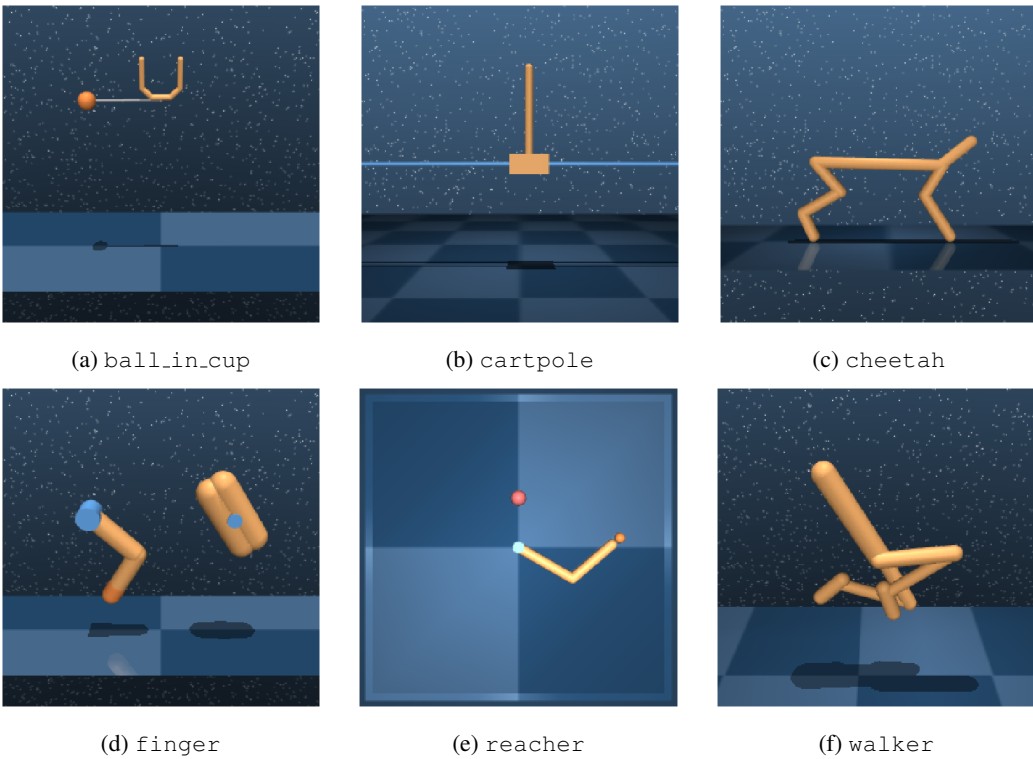

(a) `ball_in_cup`  (b) `cartpole`  (c) `cheetah`

(d) `finger`  (e) `reacher`  (f) `walker`

Figure 8: We use six domains spanning the total of twelve challenging continuous control tasks: `finger_{spin,turn_easy,turn_hard}`, `cartpole_{balance,swingup}`, `cheetah_run`, `walker_{stand,walk,run}`, `reacher_{easy,hard}`, and `ball_in_cup_catch`.

## B  HYPER PARAMETERS AND SETUP

### B.1  ACTOR AND CRITIC NETWORKS

We employ double Q-learning (van Hasselt et al., 2015) for the critic, where each Q-function is parametrized as a 3-layer MLP with `ReLU` activations after each layer except of the last. The actor is also a 3-layer MLP with `ReLU`s that outputs mean and covariance for the diagonal Gaussian that represents the policy. The hidden dimension is set to 1024 for both the critic and actor.

### B.2  ENCODER AND DECODER NETWORKS

We employ an almost identical encoder architecture as in Tassa et al. (2018), with two minor differences. Firstly, we add two more convolutional layers to the convnet trunk. Secondly, we use `ReLU` activations after each conv layer, instead of `ELU`. We employ kernels of size $3 \times 3$ with 32 channels for all the conv layers and set stride to 1 everywhere, except of the first conv layer, which has stride 2. We then take the output of the convnet and feed it into a single fully-connected layer normalized by `LayerNorm` (Ba et al., 2016). Finally, we add `tanh` nonlinearity to the 50 dimensional output of the fully-connected layer.

The actor and critic networks both have separate encoders, although we share the weights of the conv layers between them. Furthermore, only the critic optimizer is allowed to update these weights (e.g. we truncate the gradients from the actor before they propagate to the shared conv layers).

The decoder consists of one fully-connected layer that is then followed by four deconv layers. We use `ReLU` activations after each layer, except the final deconv layer that produces pixels representation. Each deconv layer has kernels of size $3 \times 3$ with 32 channels and stride 1, except of the last layer, where stride is 2.

We then combine the critic's encoder together with the decoder specified above into an autoencoder. Note, because we share conv weights between the critic's and actor's encoders, the conv layers of the actor's encoder will be also affected by reconstruction signal from the autoencoder.

### B.3  TRAINING AND EVALUATION SETUP

We first collect 1000 seed observations using a random policy. We then collect training observations by sampling actions from the current policy. We perform one training update every time we receive a new observation. In cases where we use action repeat, the number of training observations is only a fraction of the environment steps (e.g. a 1000 steps episode at action repeat 4 will only results into 250 training observations). The action repeat used for each environment is specified in Table 3, following those used by PlaNet and SLAC.

We evaluate our agent after every 10000 environment steps by computing an average episode return over 10 evaluation episodes. Instead of sampling from the Gaussian policy we take its mean during evaluation.

We preserve this setup throughout all the experiments in the paper.

| Task name | Action repeat |
|---|---|
| `cartpole_swingup` | 8 |
| `reacher_easy` | 4 |
| `cheetah_run` | 4 |
| `finger_spin` | 2 |
| `ball_in_cup_catch` | 4 |
| `walker_walk` | 2 |

Table 3: Action repeat parameter used per task, following PlaNet and SLAC.

### B.4 Weights initialization

We initialize the weight matrix of fully-connected layers with the orthogonal initialization (Saxe et al., 2013) and set the bias to be zero. For convolutional and deconvolutional layers we use delta-orthogonal initialization (Xiao et al., 2018).

### B.5 Regularization

We regularize the autoencoder network using the scheme proposed in Ghosh et al. (2019). In particular, we extend the standard reconstruction loss for a deterministic autoencoder with a $L_2$ penalty on the learned representation $z$ and add weight decay on the decoder parameters $\theta_{\text{dec}}$:

$$\mathcal{L}_{\text{RAE}} = \mathcal{L}_{\text{rec}} + \lambda_z ||z||_2^2 + \lambda_\theta ||\theta_{\text{dec}}||_2^2.$$

We set $\lambda_z = 10^{-6}$ and $\lambda_\theta = 10^{-7}$.

### B.6 Pixels preprocessing

We construct an observational input as an 3-stack of consecutive frames (Mnih et al., 2013), where each frame is a RGB rendering of size $3 \times 84 \times 84$ from the 0th camera. We then divide each pixel by 255 to scale it down to $[0, 1)$ range. For reconstruction targets we instead preprocess images by reducing bit depth to 5 bits as in Kingma & Dhariwal (2018).

### B.7 Other hyper parameters

We also provide a comprehensive overview of all the remaining hyper parameters in Table 4.

| Parameter name | Value |
|---|---|
| Replay buffer capacity | 1000000 |
| Batch size | 128 |
| Discount $\gamma$ | 0.99 |
| Optimizer | Adam |
| Critic learning rate | $10^{-3}$ |
| Critic target update frequency | 2 |
| Critic Q-function soft-update rate $\tau_Q$ | 0.01 |
| Critic encoder soft-update rate $\tau_{\text{enc}}$ | 0.05 |
| Actor learning rate | $10^{-3}$ |
| Actor update frequency | 2 |
| Actor log stddev bounds | $[-10, 2]$ |
| Autoencoder learning rate | $10^{-3}$ |
| Temperature learning rate | $10^{-4}$ |
| Temperature Adam's $\beta_1$ | 0.5 |
| Init temperature | 0.1 |

Table 4: A complete overview of used hyper parameters.

## C  ITERATIVE REPRESENTATION LEARNING WITH $\beta$-VAE

Iterative pretraining suggested in Lange & Riedmiller (2010); Finn et al. (2015) allows for faster representation learning, which consequently boosts the final performance, yet it is not sufficient enough to fully close the gap and additional modifications, such as end-to-end training, are needed. Figure 9 provides additional results for the experiment described in Section 4.3.

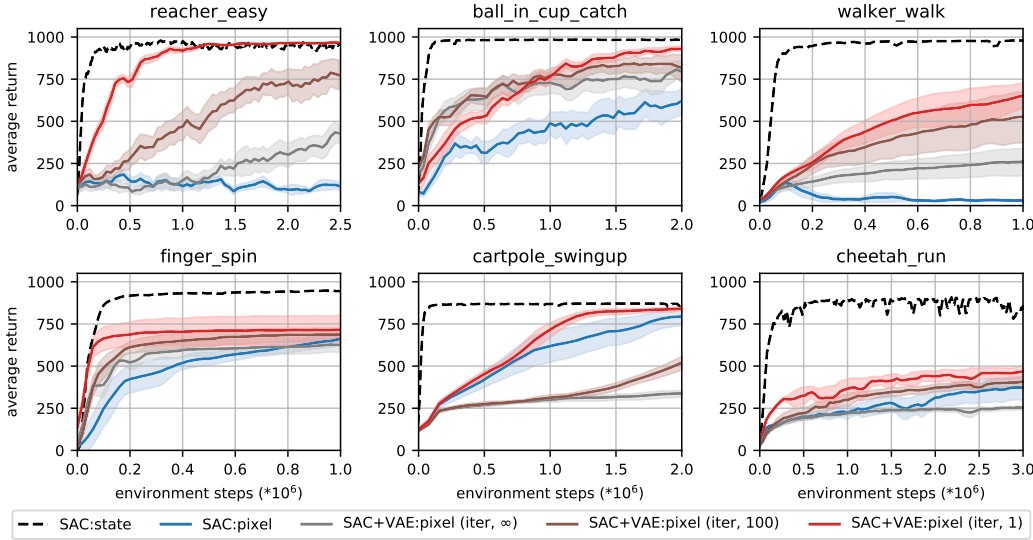

Figure 9: Separate $\beta$-VAE and policy training with no shared gradients SAC+VAE:pixel (iter, $N$), with SAC:state shown as an upper bound. $N$ refers to frequency in environment steps at which the $\beta$-VAE updates after initial pretraining. More frequent updates are beneficial for learning better representations, but cannot fully address the gap in performance.

# D    AN ATTEMPT FOR END-TO-END REPRESENTATION LEARNING WITH $\beta$-VAE

Additional results to the experiments from Section 4.4 are in Figure 10.

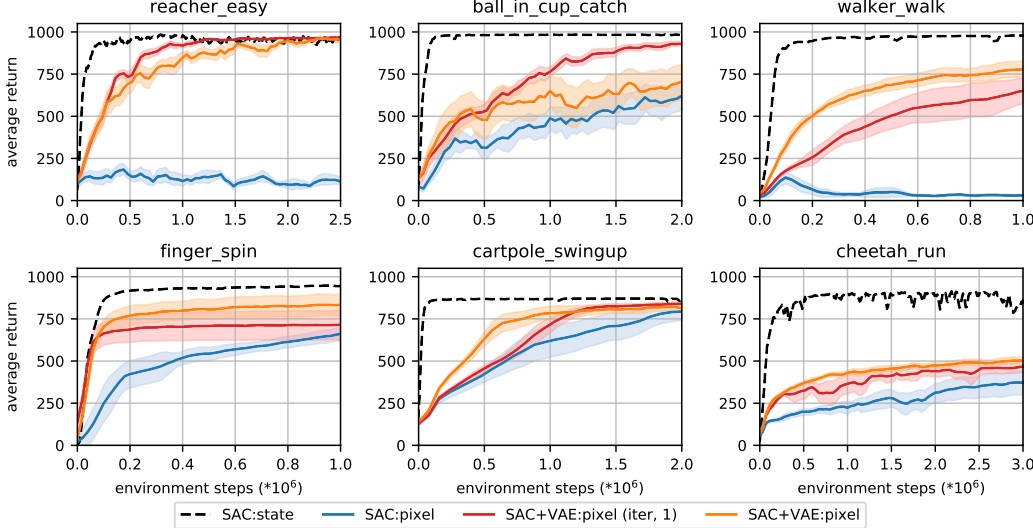

Figure 10: An unsuccessful attempt to propagate gradients from the actor-critic down to the encoder of the $\beta$-VAE to enable end-to-end off-policy training. The learning process of SAC+VAE:pixel exhibits instability together with the subpar performance comparing to the baseline SAC+VAE:pixel (iter, 1), which does not share gradients with the actor-critic.

# E CAPACITY OF THE AUTOENCODER

We also investigate various autoencoder capacities for the different tasks. Specifically, we measure the impact of changing the capacity of the convolutional trunk of the encoder and corresponding deconvolutional trunk of the decoder. Here, we maintain the shared weights across convolutional layers between the actor and critic, but modify the number of convolutional layers and number of filters per layer in Figure 11 across several environments. We find that SAC+AE is robust to various autoencoder capacities, and all architectures tried were capable of extracting the relevant features from pixel space necessary to learn a good policy. We use the same training and evaluation setup as detailed in Appendix B.3.

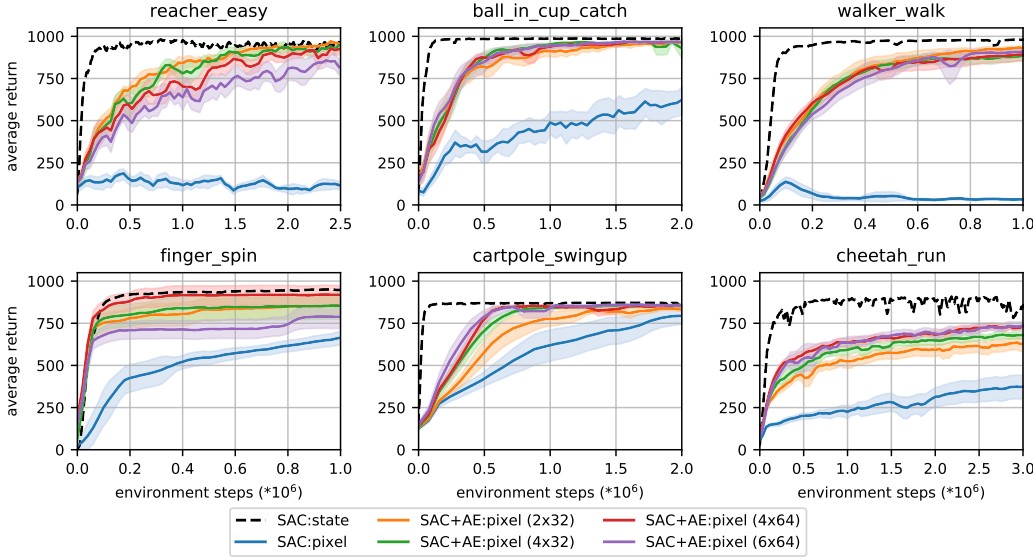

Figure 11: Different autoencoder architectures, where we vary the number of conv layers and the number of output channels in each layer in both the encoder and decoder. For example, $4 \times 32$ specifies an architecture with $4$ conv layers, each outputting $32$ channels. We observe that the difference in capacity has only limited effect on final performance.

# F    REPRESENTATION POWER OF THE ENCODER

Addition results to the experiment in Section 6.1 that demonstrates encoder's power to reconstruct proprioceptive state from image-observations are shown in Figure 12.

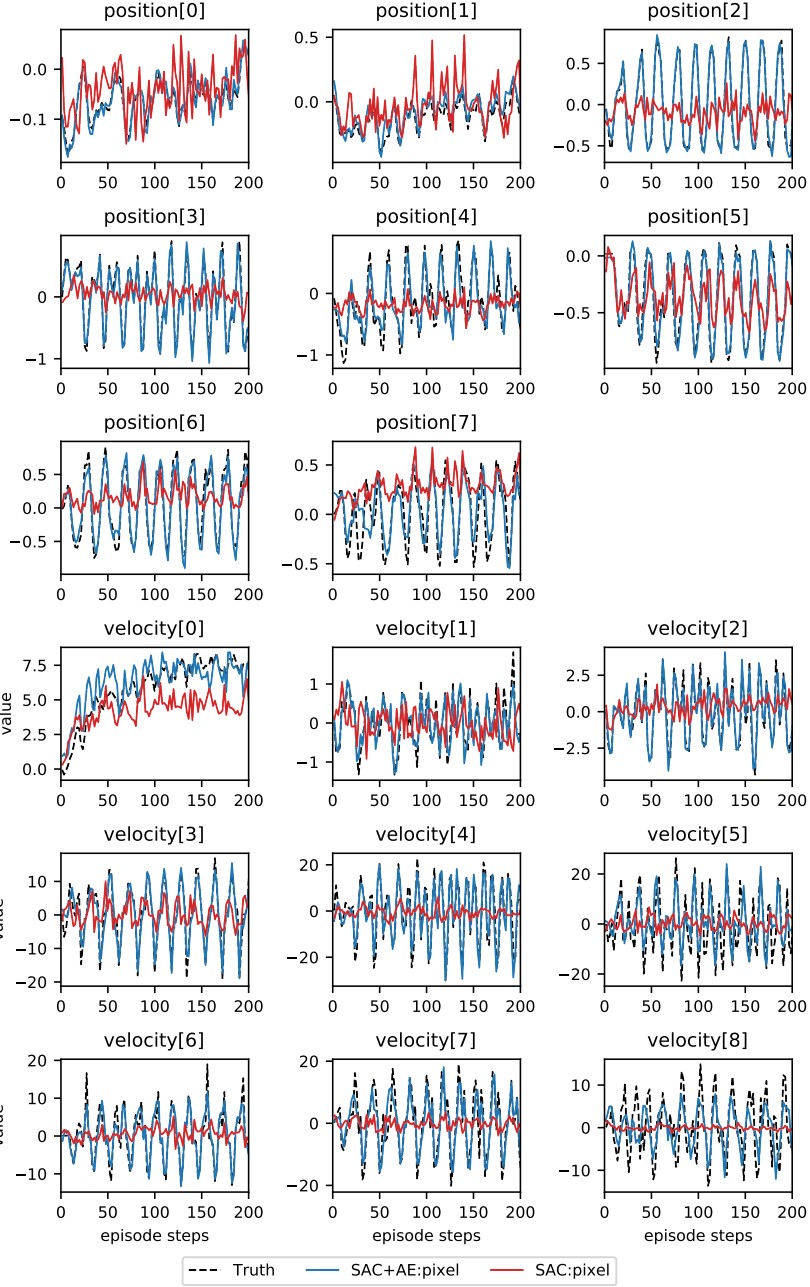

Figure 12: Linear projections of latent representation spaces learned by our method (SAC+AE:pixel) and the baseline (SAC:pixel) onto proprioceptive states. We compare ground truth value of each proprioceptive coordinate against their reconstructions for cheetah_run, and conclude that our method successfully encodes proprioceptive state information. The proprioceptive state of cheetah_run has 8 position and 9 velocity coordinates.

# G    DECODING TO PROPRIOCEPTIVE STATE

Learning from low-dimensional proprioceptive observations achieves better final performance with greater sample efficiency (see Figure 5 for comparison to pixels and Appendix J for proprioceptive baselines), therefore our intuition is to directly use these compact observations as the reconstruction targets to generate an auxiliary signal. Although, this is an unrealistic setup, given that we do not have access to proprioceptive states in practice, we use it as a tool to understand if such supervision is beneficial for representation learning and therefore can achieve good performance. We augment the observational encoder $f_{\text{enc}}$, that maps an image $\mathbf{o}_t$ into a latent vector $\mathbf{z}_t$, with a state decoder $f_{\text{state\_dec}}$, that restores the corresponding state $\mathbf{s}_t$ from the latent vector $\mathbf{z}_t$. This leads to an auxiliary objective $\mathbb{E}_{\mathbf{o}_t, \mathbf{s}_t \sim \mathcal{D}} \left[ \frac{1}{2} || f_{\text{state\_dec}}(\mathbf{z}_t) - \mathbf{s}_t ||_2^2 \right]$, where $\mathbf{z}_t = f_{\text{enc}}(\mathbf{o}_t)$. We parametrize the state decoder $f_{\text{state\_dec}}$ as a 3-layer MLP with 1024 hidden size and ReLU activations, and train it end-to-end with the actor-critic network. Such auxiliary supervision helps less than expected, and surprisingly hurts performance in ball_in_cup_catch, as seen in Figure 13. Our intuition is that such low-dimensional supervision is not able to provide the rich reconstruction error needed to fit the high-capacity convolutional encoder $f_{\text{enc}}$. We thus seek for a denser auxiliary signal and try learning latent representation spaces with pixel reconstructions.

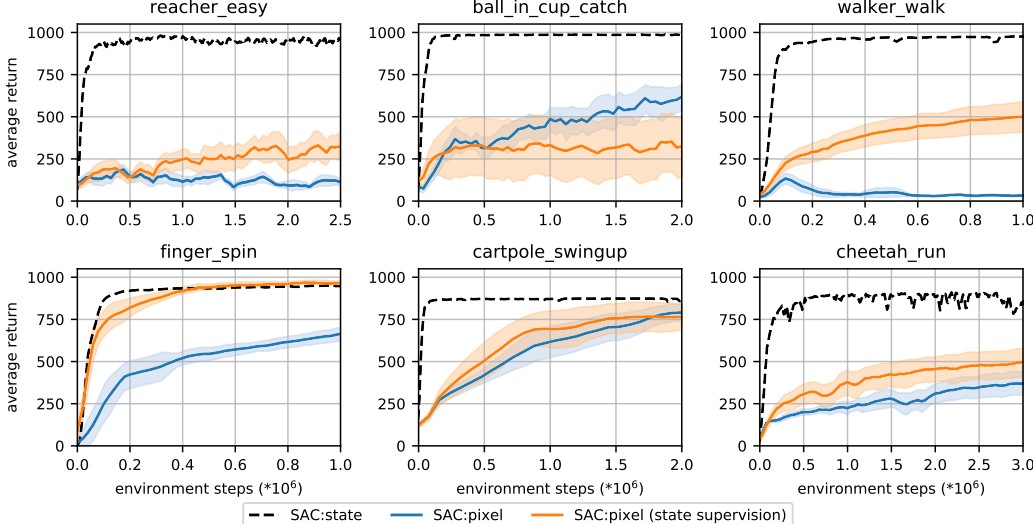

Figure 13: An auxiliary signal is provided by reconstructing a low-dimensional state from the corresponding image observation. Perhaps surprisingly, such *synthetic* supervision doesn't guarantee sufficient signal to fit the high-capacity encoder, which we infer from the suboptimal performance of SAC:pixel (state supervision) compared to SAC:pixel in ball_in_cup_catch.

# H OPTIMALITY OF LEARNED LATENT REPRESENTATION

We define the optimality of the learned latent representation as the ability of our model to extract and preserve all relevant information from the pixel observations sufficient to learn a good policy. For example, the proprioceptive state representation is clearly better than the pixel representation because we can learn a better policy. However, the differences in performance of SAC:state and SAC+AE:pixel can be attributed not only to the different observation spaces, but also the difference in data collected in the replay buffer. To decouple these attributes and determine how much information loss there is in moving from proprioceptive state to pixel images, we measure final task reward of policies learned from the same fixed replay buffer, where one is trained on proprioceptive states and the other trained on pixel observations.

We first train a SAC+AE policy until convergence and save the replay buffer that we collected during training. Importantly, in the replay buffer we store both the pixel observations and the corresponding proprioceptive states. Note that for two policies trained on the fixed replay buffer, we are operating in an off-policy regime, and thus it is possible we won't be able to train a policy that performs as well.

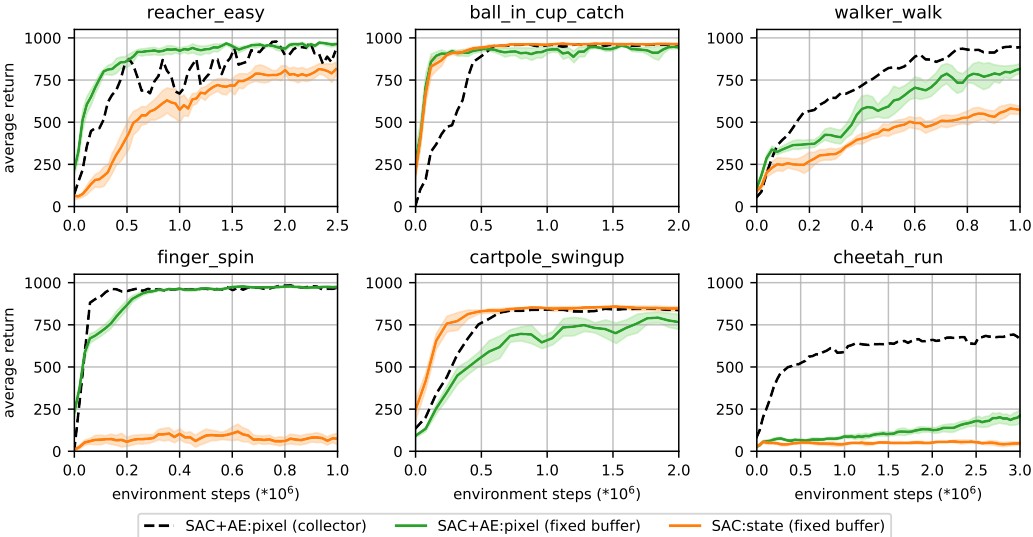

Figure 14: Training curves for the policy used to collect the buffer (SAC+AE:pixel (collector)), and the two policies learned on that buffer using proprioceptive (SAC:state (fixed buffer)) and pixel observations (SAC+AE:pixel (fixed buffer)). We see that our method actually outperforms proprioceptive observations in this setting.

In Figure 14 we find, surprisingly, that our learned latent representation outperforms proprioceptive state on a fixed buffer. This could be because the data collected in the buffer is by a policy also learned from pixel observations, and is different enough from the policy that would be learned from proprioceptive states that SAC:state underperforms in this setting.

# I  IMPORTANCE OF ACTION REPEAT

We found that repeating nominal actions several times has a significant effect on learning dynamics and final reward. Prior works (Hafner et al., 2018; Lee et al., 2019) treat action repeat as a hyper parameter to the learning algorithm, rather than a property of the target environment. Effectively, action repeat decreases the control horizon of the task and makes the control dynamics more stable. Yet, action repeat can also introduce a harmful bias, that prevents the agent from learning an optimal policy due to the injected lag. This tasks a practitioner with a problem of finding an optimal value for the action repeat hyper parameter that stabilizes training without limiting control elasticity too much.

To get more insights, we perform an ablation study, where we sweep over several choices for action repeat on multiple control tasks and compare acquired results against PlaNet (Hafner et al., 2018) with the original action repeat setting, which was also tuned per environment. We use the same setup as detailed in Appendix B.3. Specifically, we average performance over 10 random seeds, and reduce the number of training observations inverse proportionally to the action repeat value. The results are shown in Figure 15. We observe that PlaNet's choice of action repeat is not always optimal for our algorithm. For example, we can significantly improve performance of our agent on the `ball_in_cup_catch` task if instead of taking the same nominal action four times, as PlaNet suggests, we take it once or twice. The same is true on a few other environments.

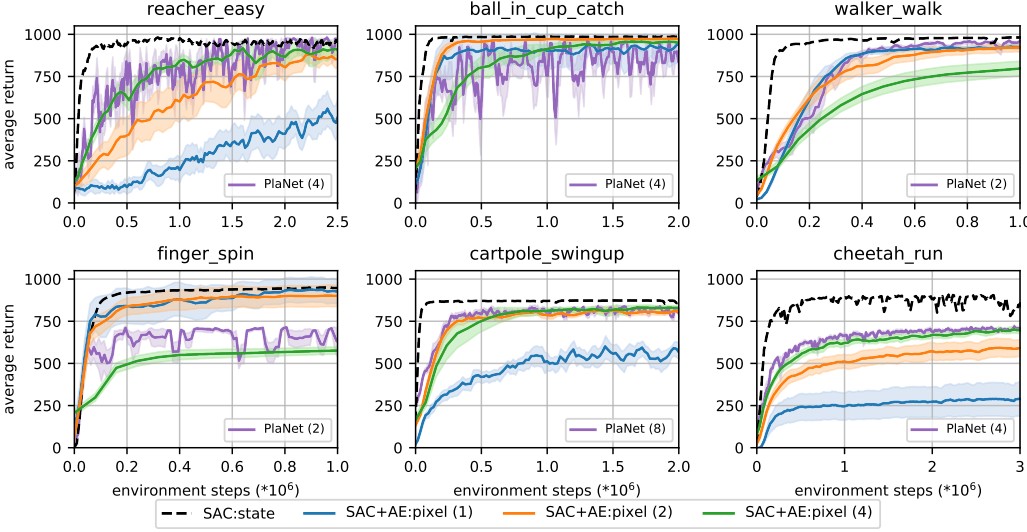

Figure 15: We study the importance of the action repeat hyper parameter on final performance. We evaluate three different settings, where the agent applies a sampled action once (SAC+AE:pixel (1)), twice (SAC+AE:pixel (2)), or four times (SAC+AE:pixel (4)). As a reference, we also plot the PlaNet (Hafner et al., 2018) results with the original action repeat setting. Action repeat has a significant effect on learning. Moreover, we note that the PlaNet's choice of hyper parameters is not always optimal for our method (e.g. it is better to apply an action only once on `walker_walk`, than taking it twice).

# J  LEARNING FROM PROPRIOCEPTIVE OBSERVATIONS

In addition to the results when an agent learns from pixels, we also provide a comprehensive comparison of several state-of-the-art continuous control algorithms that directly learn from proprioceptive states. Specifically, we consider four agents that implement SAC (Haarnoja et al., 2018), TD3 (Fujimoto et al., 2018), DDPG (Lillicrap et al., 2015), and D4PG (Barth-Maron et al., 2018). We leverage open-source implementations of TD3 and DDPG from https://github.com/sfujim/TD3, and use the reported set of optimal hyper parameters, except of the batch size, which we increase to 512, as we find it improves performance of both the algorithms. Due to lack of a publicly accessible implementation of D4PG, we take the final performance results after $10^8$ environments steps as reported

in Tassa et al. (2018). We use our own implementation of SAC together with the hyper parameters listed in Appendix B, again we increase the batch size to 512. Importantly, we keep the same set of hyper parameters across all tasks to avoid overfitting individual tasks.

For this evaluation we do not repeat actions and perform one training update per every environment step. We evaluate a policy every 10000 steps (or every 10 episodes as one episode consists of 1000 steps) by running 10 evaluation episodes and averaging corresponding returns. To assess the stability properties of each algorithm and produce reliable baselines we compute mean and std of evaluation performance over 10 random seeds. We test on twelve challenging continuous control tasks from DMC (Tassa et al., 2018), as described in Appendix A. The results are shown in Figure 16.

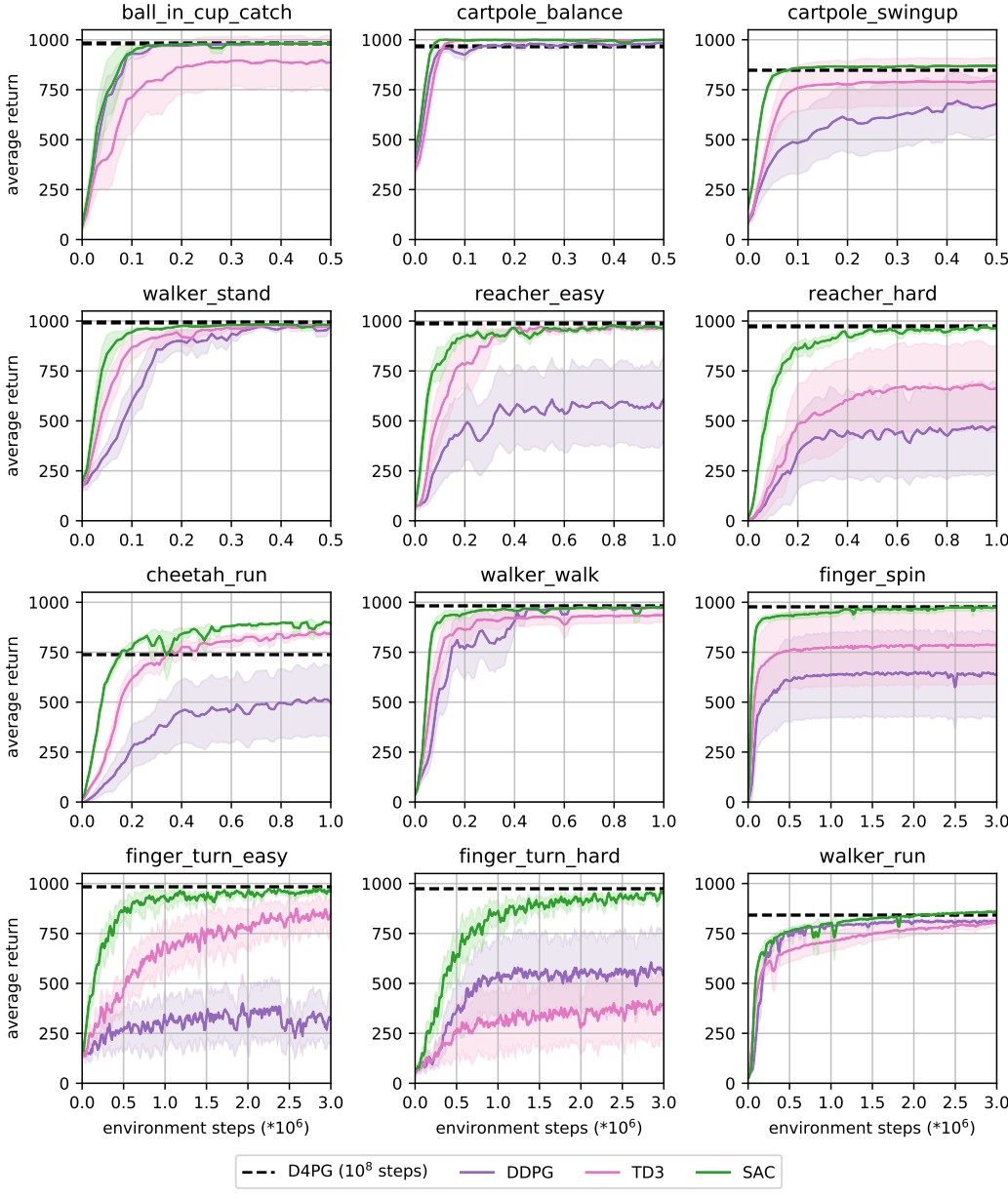

Figure 16: We benchmark SAC, TD3, DDPG, and D4PG when learning from proprioceptive states on multiple tasks of various difficulty from DMC. We run 10 random seeds for each algorithm and evaluate on 10 trajectories (except for D4PG, as its implementation is not publicly available). We then report mean and standard deviation. For D4PG we take the performance after $10^8$ environment steps reported in Tassa et al. (2018). We observe that SAC demonstrates superior performance and sample efficiency over the other methods on all of the tasks.

