# OpenReview forum: "Improving Sample Efficiency in Model-Free Reinforcement Learning from Images"
_ICLR.cc/2020/Conference — Reject_

### Official Review · AnonReviewer3 · 2019-10-08
**Official Blind Review #3**

**Rating:** 6

**Review:**

The paper aims to tackle the problem of improving sample efficiency of model-free, off-policy reinforcement learning in an image-based environment. They do so by taking SAC and adding a deterministic autoencoder, trained end-to-end with the actor and critic, with the actor and critic trained on top of the learned latent space z. They call this SAC-AE. Experiments in the DeepMind control suite demonstrate that the result models train much faster than SAC directly on the pixels, in some cases reaching close to the performance of SAC on raw state. Ablation studies demonstrate their approach is most stable with deterministic autoencoders proposed by (Ghosh et al, 2019), rather than the beta-VAE autoencoder proposed in (Nair et al, 2018), end-to-end learning of the autoencoder gives improved performance, and the encoder transfers to some similar tasks.

I thought the paper was written well, and its experiments were done quite carefully, but it was lacking on the novelty front. At a high level, the paper has many similarities with the UNREAL paper (Jaderberg et al, 2017), which is acknowledged in the related work. This paper says it differs from UNREAL because they use an off-policy algorithm, and that UNREAL's auxiliary tasks are based off real-world inductive priors.

I don't see the off-policy distinction as very relevant, because in the end, both UNREAL and SAC-AE are actor-critic algorithms (using A3C and SAC respectively). The way that SAC is used in the paper always collects data in a near on-policy manner, and UNREAL includes experience replay from a replay buffer, which introduces some off-policy nature to UNREAL as well. Therefore this doesn't feel like a strong argument.

Furthermore, although some of the auxiliary tasks in UNREAL are based off human intuition for what makes sense in those environments, they also include task-agnostic auxiliary tasks: reward prediction and pixel-level control. These do not depend on real-world inductive priors, and are shown to improve performance.

Overall, this doesn't feel like a strong enough contribution for ICLR.

More specific comments:
* Section 6.1 examines the representation power of the encoder by reconstructing proprioceptive state from the encoder. I am not sure the comparison between SAC+AE and SAC is particularly meaningful here. The predictors are learned on top of the encoder output, and in SAC+AE we would expect task information to be encoded in the learned z. But in baseline SAC, there is no reason to expect this to be true - task information is more likely to be distributed across the entire network architecture. The case for SAC+AE seems much stronger from the reward curves, rather than these plots.
* The paper argues that their approach is stable and sample-efficient, but when looking at the reward curves, it looked about as stable as SAC. Figure 3 (where they do not train the VAE end-to-end in the red curve) has a similar story. This makes me believe that any claims of added stability are more thanks to SAC, rather than proposed methods.

Edit: I would like to clarify that the rating system only provides a 3 for Weak Reject and 6 for Weak Accept. On a 1-10 scale I would rate this as a 5, I feel it is closer to Weak Accept than Weak Reject.

Edit 2: I've read the other author's comments. I'm not particularly convinced by the case for novelty, but I didn't realize that UNREAL's replay buffer was only 2k transitions instead of 1 million transitions. On reflection, I believe the main contribution here is showing that deterministic autoencoders are more reliable than stochastic ones for the RL setting, and this isn't the biggest contribution, but it's enough to make me update to weak accept.

**Experience Assessment:**

I have published in this field for several years.

**Review Assessment: Checking Correctness Of Derivations And Theory:**

I assessed the sensibility of the derivations and theory.

**Review Assessment: Checking Correctness Of Experiments:**

I carefully checked the experiments.

**Review Assessment: Thoroughness In Paper Reading:**

I read the paper at least twice and used my best judgement in assessing the paper.

---

### Official Review · AnonReviewer2 · 2019-10-08
**Official Blind Review #2**

**Rating:** 6

**Review:**

This work presents a simple method for model-free RL from image observations. The key component of the method is the addition of an autoencoder that is trained jointly with the policy and value function, in contrast to previous methods which separate feature learning from policy learning. Another important modification is the use of a deterministic regularized autoencoder instead of a stochastic variational autoencoder. The method is evaluated a variety of control tasks, and shows strong performance when compared to a number of state-of-the-art model-based and model-free methods for RL with image observations.

The paper is well written and provides a very clear description of the method. The approach is fairly simple and appears to be effective for a suite of challenging tasks. RL from images remains a very challenging problem, and the approach outlined in this work could have a significant impact on the community. The experiments are also thorough and well thought out, and the release of the source code is much appreciated. While the overall novelty is a bit limited, this could be a case where details matter, and insights provided by this work can be valuable for the community. For these reasons, I would like to recommend acceptance.

There is mention of SLAC as a model-based algorithm. This is not entirely accurate. SLAC does learn a dynamics model as means of acquiring a latent state-representation, but this model is not used to train the policy or for planning at runtime. The policy in SLAC is trained in a model-free manner.


**Experience Assessment:**

I have published one or two papers in this area.

**Review Assessment: Checking Correctness Of Derivations And Theory:**

N/A

**Review Assessment: Checking Correctness Of Experiments:**

I carefully checked the experiments.

**Review Assessment: Thoroughness In Paper Reading:**

I read the paper thoroughly.

---

### Official Review · AnonReviewer1 · 2019-10-17
**Official Blind Review #1**

**Rating:** 3

**Review:**

Summary

This paper proposes an approach to make the model-free state-of-the-art soft actor-critic (SAC) algorithm for proprioceptive state spaces sample-efficient in higher-dimensional visual state spaces. To this end, an encoder-decoder structure to minimize image reconstruction loss is added to SAC's learning objectives. Importantly, the encoder is shared between the encoder-decoder architecture, the critic and the policy. Furthermore, Q-critic updates backpropagate through the encoder such that encoder weights need to trade off image reconstruction and critic learning. The approach is evaluated on six tasks from the DeepMind control suite and compared against proprioceptive SAC, pixel-based SAC, D4PG as well as to the model-based baselines PlaNet and SLAC. The proposed method seems to achieve results competitive with the model-based baselines and significantly improves over raw pixel-based SAC. Further ablation studies are presented to investigate the information capacity of the learned latent representation and generalization to unseen tasks.

Quality

Since this is a paper with a strong practical focus, the quality needs to be judged based on the experiments. The quality of those are good in terms of the number of environments, baselines, benchmarks and seeds. I also liked the ablation studies to investigate latent representations and generalization to new tasks.

Clarity

The paper is very clearly written and easy to follow.

Originality

Unfortunately, the originality is very low. Combining reinforcement learning with auxiliary objectives is not novel and has been studied in the Atari domain (discrete actions) as noted by the authors, see Jaderberg et al., ICLR, 2017 and Shelhamer et al., arXiv, 2017. The conceptual idea of using a reconstruction loss for images as auxiliary objective is not novel either and has been presented in earlier work already, see Shelhamer et al. The idea of sharing parameters between RL and auxiliary components is also not novel, see Jaderberg et al. One citation that is conceptually very similar to the authors' work is missing: 'Felix Leibfried and Peter Vrancx, Model-based regularization for deep reinforcement learning with transcoder networks. In NeurIPS Deep Reinforcement Learning Workshop, 2018'. The former work combines Q-value learning with auxiliary losses for learning an environment model end to end (with a reconstruction loss for the next state) in the domain of Atari.

Significance

The significance is minor to low. The fact that the authors investigate auxiliary losses in continuous-action domains has minor significance. But all in all, the paper might be better suited for a workshop rather than the main track of ICLR.

Minor Details

On page 3, first equation (not numbered), there is an average over s_{t+1} missing because of the reward definition used by the authors?

Update

I read the other reviews and the authors' response. I still feel that the novelty of the work is very limited and the authors' response to lacking novelty does not convince me. However, in light of the strong experimental analysis, I feel in hindsight that a score of 1 from my side was too harsh. I therefore increase my score to 3, but I do still believe that the paper is better suited as a workshop contribution.

**Experience Assessment:**

I have published in this field for several years.

**Review Assessment: Checking Correctness Of Derivations And Theory:**

I assessed the sensibility of the derivations and theory.

**Review Assessment: Checking Correctness Of Experiments:**

I assessed the sensibility of the experiments.

**Review Assessment: Thoroughness In Paper Reading:**

I read the paper at least twice and used my best judgement in assessing the paper.

---

### Author Response · Authors · 2019-11-08
**Review response -- thanks for the feedback! [Part 1 of 2]**

We thank the reviewers for their comments. In particular, we were gratified by R2’s positive observations: “The approach is fairly simple and appears to be effective for a suite of challenging tasks”; “this work could have a significant impact on the community” and “The experiments are also thorough and well thought out”. We also appreciate that the reviewers find our paper to be well written and easy to follow.


R1, R3: Lack of novelty.
We respectfully disagree with R1 and R3’s concerns. As we discuss in the paper, auxiliary objectives, including reconstruction loss, have certainly been used before in RL. However, the empirical performance of previous approaches is dramatically worse than our approach in Mujoco settings. This discrepancy is noted by R2, who correctly recognizes this as “a case where details matter”.

We are thus concerned that R1 and R3 may not fully appreciate this aspect of our paper. This concern is bolstered by R1’s view that our approach is that same as that of [Shelhamer et al.’17], when in fact this work vividly illustrates how differing “details” lead to very different experimental outcomes.

Like our paper, [Shelhamer et al.’17] explored an auxiliary reconstruction loss (amongst others). But their training setup differs from ours in a variety of ways which turn out to be crucial to performance. In section 4.2, the authors note that “Reconstruction by VAE is mostly harmful” and that “The VAE even diverges for several environments” when trained stage-wise (section 4.3). Possibly because of this, the VAE is absent from their end-to-end training experiments (section 4.5). Thus due to an incorrect training protocol the authors (erroneously) conclude that an input reconstruction auxiliary loss is not effective. By contrast, our paper devises an *effective* training protocol for an input reconstruction auxiliary loss and shows that it is key to obtaining SOTA-comparable performance.


R3: Relation to UNREAL [Jaderberg et al.’17] -- on vs off-policy.
While UNREAL uses a replay buffer to train the critic and auxiliary models in off-policy fashion, the actor itself is trained on-policy via policy gradients (first paragraph in section 3.4).

The quasi off-policy nature of training in UNREAL is further demonstrated by the manner in which the replay buffer is used. The replay buffer in UNREAL “stores the most recent 2k observations, actions, and rewards taken by the base agent” (appendix B), a small fraction of the 25M transitions experienced in training. By contrast, in our true off-policy setting, the replay buffer stores 1M transitions (which usually is the entirety of all training transitions) and samples are drawn *uniformly* from *all* accumulated experiences. Our approach is able to achieve superior sample efficiency, despite the out-of-distribution nature of the replay buffer samples, thanks to the auxiliary reconstruction loss.


R3: Auxiliary tasks in UNREAL [Jaderberg et al.’17].
As with [Shelhamer et al.’17], UNREAL explores reconstruction loss as an auxiliary task for A3C, but find that it performs worse than A3C alone (see Fig. 5(left) in [Jaderberg et al.’17]). In contrast, with our approach we obtain dramatic performance gains by adding a generic input reconstruction loss.

Of the other auxiliary tasks considered in UNREAL, Pixel Control is the most effective. However, in maximizing changes in local patches, it imposes strong inductive biases that assume that dramatically changing pixel values and textures are correlated with good exploration and reward. The other generic auxiliary tasks (reward prediction and value prediction) only provide marginal improvements (see Fig. 3(top left) in [Jaderberg et al.’17]) and may require specific reward structure (e.g. dense reward).


R3: Stability of training is due to SAC.
We disagree. Augmenting SAC with a VAE (very similar conceptually to the RAE used in our approach) makes it highly unstable and sensitive to the choice of \beta (see https://drive.google.com/open?id=1qYeiPXYl0iEmJYImZxDNhgjtNrDWtlfL ). Thus seems implausible that SAC is the main source of stability in our method. Furthermore,  [Shelhamer et al.’17] combine VAE’s with several RL algorithms and also find that it makes them unstable.


R1, R3: Importance of simplicity.
We respectfully feel that this aspect has been overlooked by R1 and R3. Our method matches the performance of current SOTA methods, while being far simpler (both from an implementation and conceptual standpoint). For many reasons, this should make our approach preferable, not least of which is reproducibility. Indeed, our approach is straightforward to reimplement, unlike many RL algorithms.  We also support our submission with a compact and easy to understand PyTorch implementation.

---

### Author Response · Authors · 2019-11-08
**Review response -- thanks for the feedback! [Part 2 of 2]**

Minor comments:

R1: Achieve results competitive with the model-based baselines.
Please note that these “baselines” are the current state-of-art-methods.


R1: Typo in the equation 1.
Thank you for spotting this typo, we will update the paper.


R1: Missing reference.
We will update our paper with the missing reference, thank you for pointing this out.


R2: SLAC is not being model-based.
We concur with the reviewer that SLAC does not use transitions sampled from the model for training and perhaps should be labeled as a model-free method. Our primary motivation was to showcase that SLAC trains a complicated latent model for dynamics that enjoys a lot of auxiliary supervision. In contrast, our method achieve competitive performance while being significantly simpler. We will update the wording in the paper to make the classification clearer.


R3: Representation power.
We would like to point out that the encoder attributes for 90% of weights of our agent. The Q-function and policy networks are just small 3 layers MLPs on top of the encoder. Thus, we believe that the encoder should encapsulate some meaningful representations of internal states and our experiment is an adequate way to measure the amount of captured information.

---

### Decision · Program_Chairs · 2019-12-19

**Decision:**

Reject

**Comment:**

The paper investigates how sample efficiency of image based model-free RL can be improved  by including an image reconstruction loss as an auxiliary task and applies it to soft actor-critic. The method is demonstrated to yield a substantial improvement compared to SAC learned directly from pixels, and comparable performance to other prior works, such as SLAC and PlaNet, but with a simpler learning setup. The reviewers generally appreciate the clarity of presentation and good experimental evaluation. However, all reviewers raise concerns regarding limited novelty, as auxiliary losses for RL have been studied before, and the contribution is mainly in the design choices of the implementation. In this view, and given that the results are on a par with SOTA, the contribution of this paper seems too incremental for publishing in this venue, and I’m recommending rejection.